# Homotopy Smoothing for Non-Smooth Problems with Lower Complexity than $O(1/\epsilon)$

**Yi Xu[†*], Yan Yan[‡*], Qihang Lin[♮], Tianbao Yang ✉[†]**

[†] Department of Computer Science, University of Iowa, Iowa City, IA 52242
[‡] QCIS, University of Technology Sydney, NSW 2007, Australia
[♮] Department of Management Sciences, University of Iowa, Iowa City, IA 52242
{yi-xu, qihang-lin, tianbao-yang}@uiowa.edu, yan.yan-3@student.uts.edu.au

## Abstract

In this paper, we develop a novel **ho**moto**p**y **s**moothing (HOPS) algorithm for solving a family of non-smooth problems that is composed of a non-smooth term with an explicit max-structure and a smooth term or a simple non-smooth term whose proximal mapping is easy to compute. The best known iteration complexity for solving such non-smooth optimization problems is $O(1/\epsilon)$ without any assumption on the strong convexity. In this work, we will show that the proposed HOPS achieved a lower iteration complexity of $\tilde{O}(1/\epsilon^{1-\theta})$ [1]with $\theta \in (0, 1]$ capturing the local sharpness of the objective function around the optimal solutions. To the best of our knowledge, this is the lowest iteration complexity achieved so far for the considered non-smooth optimization problems without strong convexity assumption. The HOPS algorithm employs Nesterov's smoothing technique and Nesterov's accelerated gradient method and runs in stages, which gradually decreases the smoothing parameter in a stage-wise manner until it yields a sufficiently good approximation of the original function. We show that HOPS enjoys a linear convergence for many well-known non-smooth problems (e.g., empirical risk minimization with a piece-wise linear loss function and $\ell_1$ norm regularizer, finding a point in a polyhedron, cone programming, etc). Experimental results verify the effectiveness of HOPS in comparison with Nesterov's smoothing algorithm and the primal-dual style of first-order methods.

## 1 Introduction

In this paper, we consider the following optimization problem:

$$\min_{x \in \Omega_1} F(x) \triangleq f(x) + g(x) \tag{1}$$

where $g(x)$ is a convex (but not necessarily smooth) function, $\Omega_1$ is a closed convex set and $f(x)$ is a convex but non-smooth function which can be explicitly written as

$$f(x) = \max_{u \in \Omega_2} \langle Ax, u \rangle - \phi(u) \tag{2}$$

where $\Omega_2 \subset \mathbb{R}^m$ is a closed convex bounded set, $A \in \mathbb{R}^{m \times d}$ and $\phi(u)$ is a convex function, and $\langle \cdot, \cdot \rangle$ is scalar product. This family of non-smooth optimization problems has applications in numerous domains, e.g., machine learning and statistics [7], image processing [6], cone programming [11], and etc. Several first-order methods have been developed for solving such non-smooth optimization

---

[*]The first two authors make equal contributions. The work of Y. Yan was done when he was a visiting student at Department of Computer Science of the University of Iowa.

[1]$\tilde{O}()$ suppresses a logarithmic factor.

problems including the primal-dual methods [15, 6], Nesterov's smoothing algorithm [16] [2], and they can achieve $O(1/\epsilon)$ iteration complexity for finding an $\epsilon$-optimal solution, which is faster than the corresponding black-box lower complexity bounds by an order of magnitude.

In this paper, we propose a novel homotopy smoothing (HOPS) algorithm for solving the problem in (1) that achieves a lower iteration complexity than $O(1/\epsilon)$. In particular, the iteration complexity of HOPS is given by $\tilde{O}(1/\epsilon^{1-\theta})$, where $\theta \in (0, 1]$ captures the local sharpness (defined shortly) of the objective function around the optimal solutions. The proposed HOPS algorithm builds on the Nesterov's smoothing technique, i.e., approximating the non-smooth function $f(x)$ by a smooth function and optimizing the smoothed function to a desired accuracy level.

The striking difference between HOPS and Nesterov's smoothing algorithm is that Nesterov uses a fixed small smoothing parameter that renders a sufficiently accurate approximation of the non-smooth function $f(x)$, while HOPS adopts a homotopy strategy for setting the value of the smoothing parameter. It starts from a relatively large smoothing parameter and gradually decreases the smoothing parameter in a stage-wise manner until the smoothing parameter reaches a level that gives a sufficiently good approximation of the non-smooth objective function. The benefit of using a homotopy strategy is that a larger smoothing parameter yields a smaller smoothness constant and hence a lower iteration complexity for smoothed problems in earlier stages. For smoothed problems in later stages with larger smoothness constants, warm-start can help reduce the number of iterations to converge. As a result, solving a series of smoothed approximations with a smoothing parameter from large to small and with warm-start is faster than solving one smoothed approximation with a very small smoothing parameter. To the best of our knowledge, **this is the first work** that rigorously analyzes such a homotopy smoothing algorithm and establishes its theoretical guarantee on lower iteration complexities. The keys to our analysis of lower iteration complexity are (i) to leverage a **global error inequality** (Lemma 1) [21] that bounds the distance of a solution to the $\epsilon$ sublevel set by a multiple of the functional distance; and (ii) to explore a local error bound condition to bound the multiplicative factor.

## 2   Related Work

In this section, we review some related work for solving the considered family of non-smooth optimization problems.

In the seminal paper by Nesterov [16], he proposed a smoothing technique for a family of structured non-smooth optimization problems as in (1) with $g(x)$ being a smooth function and $f(x)$ given in (2). By adding a strongly convex prox function in terms of $u$ with a smoothing parameter $\mu$ into the definition of $f(x)$, one can obtain a smoothed approximation of the original objective function. Then he developed an accelerated gradient method with an $O(1/t^2)$ convergence rate for the smoothed objective function with $t$ being the number of iterations, which implies an $O(1/t)$ convergence rate for the original objective function by setting $\mu \approx c/t$ with $c$ being a constant. The smoothing technique has been exploited to solving problems in machine learning, statistics, cone programming [7, 11, 24].

The primal-dual style of first-order methods treat the problem as a convex-concave minimization problem, i.e.,

$$\min_{x \in \Omega_1} \max_{u \in \Omega_2} g(x) + \langle Ax, u \rangle - \phi(u)$$

Nemirovski [15] proposed a mirror prox method, which has a convergence rate of $O(1/t)$ by assuming that both $g(x)$ and $\phi(u)$ are smooth functions. Chambolle & Pock [6] designed first-order primal-dual algorithms, which tackle $g(x)$ and $\phi(u)$ using proximal mapping and achieve the same convergence rate of $O(1/t)$ without assuming smoothness of $g(x)$ and $\phi(u)$. When $g(x)$ or $\phi(u)$ is strongly convex, their algorithms achieve $O(1/t^2)$ convergence rate. The effectiveness of their algorithms was demonstrated on imaging problems. Recently, the primal-dual style of first-order methods have been employed to solve non-smooth optimization problems in machine learning where both the loss function and the regularizer are non-smooth [22]. Lan et al. [11] also considered Nemirovski's prox method for solving cone programming problems.

The key condition for us to develop an improved convergence is closely related to local error bounds (LEB) [17] and more generally the Kurdyka-Łojasiewicz property [12, 4]. The LEB characterizes

the relationship between the distance of a local solution to the optimal set and the optimality gap of the solution in terms of objective value. The Kurdyka-Łojasiewicz property characterizes that property of a function that whether it can be made "sharp" by some transformation. Recently, these conditions/properties have been explored for feasible descent methods [13], non-smooth optimization [8], gradient and subgradient methods [10, 21]. **It is notable** that our local error bound condition is different from the one used in [13, 25] which bounds the distance of a point to the optimal set by the norm of the projected or proximal gradient at that point instead of the functional distance, consequentially it requires some smoothness assumption about the objective function. By contrast, the local error bound condition in this paper covers a much broad family of functions and thus it is more general. Recent work [14, 23] have shown that the error bound in [13, 25] is a special case of our considered error bound with $\theta = 1/2$. Two mostly related work leveraging a similar error bound to ours are discussed in order. Gilpin et al. [8] considered the two-person zero-sum games, which is a special case of (1) with $g(x)$ and $\phi(u)$ being zeros and $\Omega_1$ and $\Omega_2$ being polytopes. The present work is a non-trivial generalization of their work that leads to improved convergence for a much broader family of non-smooth optimization problems. In particular, their result is just a special case of our result when the constant $\theta$ that captures the local sharpness is one for problems whose epigraph is a polytope. Recently, Yang & Lin [21] proposed a restarted subgradient method by exploring the local error bound condition or more generally the Kurdyka-Łojasiewicz property, resulting in an $\tilde{O}(1/\epsilon^{2(1-\theta)})$ iteration complexity with the same constant of $\theta$. In contrast, our result is an improved iteration complexity of $\tilde{O}(1/\epsilon^{1-\theta})$.

It is worth emphasizing that the proposed homotopy smoothing technique is different from recently proposed homotopy methods for sparse learning (e.g., $\ell_1$ regularized least-squares problem [20]), though a homotopy strategy on an involved parameter is also employed to boost the convergence. In particular, the involved parameter in the homotopy methods for sparse learning is the regularization parameter before the $\ell_1$ regularization, while the parameter in the present work is the introduced smoothing parameter. In addition, the benefit of starting from a relatively large regularization parameter in sparse learning is the sparsity of the solution, which makes it possible to explore the restricted strong convexity for proving faster convergence. We do not make such assumption of the data and we are mostly interested in that when both $f(x)$ and $g(x)$ are non-smooth. Finally, we note that a similar homotopy (a.k.a continuation) strategy is employed in Nesterov's smoothing algorithm for solving an $\ell_1$ norm minimization problem subject to a constraint for recovering a sparse solution [3]. However, we would like to draw readers' attention to that they did not provide any theoretical guarantee on the iteration complexity of the homotopy strategy and consequentially their implementation is ad-hoc without guidance from theory. More importantly, our developed algorithms and theory apply to a much broader family of problems.

## 3 Preliminaries

We present some preliminaries in this section. Let $\|x\|$ denote the Euclidean norm on the primal variable $x$. A function $h(x)$ is $L$-smooth in terms of $\|\cdot\|$, if $\|\nabla h(x) - \nabla h(y)\| \leq L\|x - y\|$. Let $\|u\|_+$ denote a norm on the dual variable, which is not necessarily the Euclidean norm. Denote by $\omega_+(u)$ a 1-strongly convex function of $u$ in terms of $\|\cdot\|_+$.

For the optimization problem in (1), we let $\Omega_*, F_*$ denote the set of optimal solutions and optimal value, respectively, and make the following assumption throughout the paper.

**Assumption 1.** *For a convex minimization problem (1), we assume (i) there exist $x_0 \in \Omega_1$ and $\epsilon_0 \geq 0$ such that $F(x_0) - \min_{x \in \Omega_1} F(x) \leq \epsilon_0$; (ii) $f(x)$ is characterized as in (2), where $\phi(u)$ is a convex function; (iii) There exists a constant $D$ such that $\max_{u \in \Omega_2} \omega_+(u) \leq D^2/2$; (iv) $\Omega_*$ is a non-empty convex compact set.*

Note that: 1) Assumption 1(i) assumes that the objective function is lower bounded; 2) Assumption 1(iii) assumes that $\Omega_2$ is a bounded set, which is also required in [16].

In addition, for brevity we assume that $g(x)$ is simple enough [3] such that the proximal mapping defined below is easy to compute similar to [6]:

$$P_{\lambda g}(x) = \min_{z \in \Omega_1} \frac{1}{2}\|z - x\|^2 + \lambda g(z) \qquad (3)$$

Relying on the proximal mapping, the key updates in the optimization algorithms presented below take the following form:

$$\Pi^c_{v,\lambda g}(x) = \arg\min_{z \in \Omega_1} \frac{c}{2}\|z - x\|^2 + \langle v, z \rangle + \lambda g(z) \tag{4}$$

For any $x \in \Omega_1$, let $x^*$ denote the closest optimal solution in $\Omega_*$ to $x$ measured in terms of $\|\cdot\|$, i.e., $x^* = \arg\min_{z \in \Omega_*} \|z - x\|^2$, which is unique because $\Omega_*$ is a non-empty convex compact set We denote by $\mathcal{L}_\epsilon$ the $\epsilon$-level set of $F(x)$ and by $\mathcal{S}_\epsilon$ the $\epsilon$-sublevel set of $F(x)$, respectively, i.e.,

$$\mathcal{L}_\epsilon = \{x \in \Omega_1 : F(x) = F_* + \epsilon\}, \quad \mathcal{S}_\epsilon = \{x \in \Omega_1 : F(x) \leq F_* + \epsilon\}$$

It follows from [18] (Corollary 8.7.1) that the sublevel set $\mathcal{S}_\epsilon$ is bounded for any $\epsilon \geq 0$ and so as the level set $\mathcal{L}_\epsilon$ due to that $\Omega_*$ is bounded. Define $dist(\mathcal{L}_\epsilon, \Omega_*)$ to be the maximum distance of points on the level set $\mathcal{L}_\epsilon$ to the optimal set $\Omega_*$, i.e.,

$$dist(\mathcal{L}_\epsilon, \Omega_*) = \max_{x \in \mathcal{L}_\epsilon} \left[ dist(x, \Omega_*) \triangleq \min_{z \in \Omega_*} \|x - z\| \right]. \tag{5}$$

Due to that $\mathcal{L}_\epsilon$ and $\Omega_*$ are bounded, $dist(\mathcal{L}_\epsilon, \Omega_*)$ is also bounded. Let $x^\dagger_\epsilon$ denote the closest point in the $\epsilon$-sublevel set to $x$, i.e.,

$$x^\dagger_\epsilon = \arg\min_{z \in \mathcal{S}_\epsilon} \|z - x\|^2 \tag{6}$$

It is easy to show that $x^\dagger_\epsilon \in \mathcal{L}_\epsilon$ when $x \notin \mathcal{S}_\epsilon$ (using the KKT condition).

## 4 Homotopy Smoothing

### 4.1 Nesterov's Smoothing

We first present the Nesterov's smoothing technique and accelerated proximal gradient methods for solving the smoothed problem due to that the proposed algorithm builds upon these techniques. The idea of smoothing is to construct a smooth function $f_\mu(x)$ that well approximates $f(x)$. Nesterov considered the following function

$$f_\mu(x) = \max_{u \in \Omega_2} \langle Ax, u \rangle - \phi(u) - \mu\omega_+(u)$$

It was shown in [16] that $f_\mu(x)$ is smooth w.r.t $\|\cdot\|$ and its smoothness parameter is given by $L_\mu = \frac{1}{\mu}\|A\|^2$ where $\|A\|$ is defined by $\|A\| = \max_{\|x\| \leq 1} \max_{\|u\|_+ \leq 1} \langle Ax, u \rangle$. Denote by

$$u_\mu(x) = \arg\max_{u \in \Omega_2} \langle Ax, u \rangle - \phi(u) - \mu\omega_+(u)$$

The gradient of $f_\mu(x)$ is computed by $\nabla f_\mu(x) = A^\top u_\mu(x)$. Then

$$f_\mu(x) \leq f(x) \leq f_\mu(x) + \mu D^2/2 \tag{7}$$

From the inequality above, we can see that when $\mu$ is very small, $f_\mu(x)$ gives a good approximation of $f(x)$. This motivates us to solve the following composite optimization problem

$$\min_{x \in \Omega_1} F_\mu(x) \triangleq f_\mu(x) + g(x)$$

Many works have studied such an optimization problem [2, 19] and the best convergence rate is given by $O(L_\mu/t^2)$, where $t$ is the total number of iterations. We present a variant of accelerated proximal gradient (APG) methods in Algorithm 1 that works even with $\|x\|$ replaced with a general norm as long as its square is strongly convex. We make several remarks about Algorithm 1: (i) the variant here is similar to Algorithm 3 in [19] and the algorithm proposed in [16] except that the prox function $d(x)$ is replaced by $\|x - x_0\|^2/2$ in updating the sequence of $z_k$, which is assumed to be $\sigma_1$-strongly convex w.r.t $\|\cdot\|$; (ii) If $\|\cdot\|$ is simply the Euclidean norm, a simplified algorithm with only one update in (4) can be used (e.g., FISTA [2]); (iii) if $L_\mu$ is difficult to compute, we can use the backtracking trick (see [2, 19]).

The following theorem states the convergence result for APG.

**Theorem 2.** *([19]) Let* $\theta_k = \frac{2}{k+2}$, $\alpha_k = \frac{2}{k+1}, k \geq 0$ *or* $\alpha_{k+1} = \theta_{k+1} = \frac{\sqrt{\theta_k^4 + 4\theta_k^2} - \theta_k^2}{2}, k \geq 0$. *For any* $x \in \Omega_1$, *we have*

$$F_\mu(x_t) - F_\mu(x) \leq \frac{2L_\mu\|x - x_0\|^2}{t^2} \tag{8}$$

**Algorithm 1** An Accelerated Proximal Gradient Method: $\text{APG}(x_0, t, L_\mu)$

---

1: **Input**: the number of iterations $t$, the initial solution $x_0$, and the smoothness constant $L_\mu$
2: Let $\theta_0 = 1$, $V_{-1} = 0$, $\Gamma_{-1} = 0$, $z_0 = x_0$
3: Let $\alpha_k$ and $\theta_k$ be two sequences given in Theorem 2.
4: **for** $k = 0, \ldots, t-1$ **do**
5:     Compute $y_k = (1 - \theta_k)x_k + \theta_k z_k$
6:     Compute $v_k = \nabla f_\mu(y_k)$, $V_k = V_{k-1} + \frac{v_k}{\alpha_k}$, and $\Gamma_k = \Gamma_{k-1} + \frac{1}{\alpha_k}$
7:     Compute $z_{k+1} = \Pi_{V_k, \Gamma_k g}^{L_\mu / \sigma_1}(x_0)$ and $x_{k+1} = \Pi_{v_k, g}^{L_\mu}(y_k)$
8: **end for**
9: **Output**: $x_t$

---

Combining the above convergence result with the relation in (7), we can establish the iteration complexity of Nesterov's smoothing algorithm for solving the original problem (1).

**Corollary 3.** *For any $x \in \Omega_1$, we have*

$$F(x_t) - F(x) \le \mu D^2/2 + \frac{2L_\mu \|x - x_0\|^2}{t^2} \tag{9}$$

*In particular in order to have $F(x_t) \le F_* + \epsilon$, it suffices to set $\mu \le \frac{\epsilon}{D^2}$ and $t \ge \frac{2D\|A\|\|x_0 - x_*\|}{\epsilon}$, where $x_*$ is an optimal solution to (1).*

### 4.2   Homotopy Smoothing

From the convergence result in (9), we can see that in order to obtain a very accurate solution, we have to set $\mu$ - the smoothing parameter - to be a very small value, which will cause the blow-up of the second term because $L_\mu \propto 1/\mu$. On the other hand, if $\mu$ is set to be a relatively large value, then $t$ can be set to be a relatively small value to match the first term in the R.H.S. of (9), which may lead to a not sufficiently accurate solution. It seems that the $O(1/\epsilon)$ is unbeatable. However, if we adopt a homotopy strategy, i.e., starting from a relatively large value $\mu$ and optimizing the smoothed function with a certain number of iterations $t$ such that the second term in (9) matches the first term, which will give $F(x_t) - F(x_*) \le O(\mu)$. Then we can reduce the value of $\mu$ by a constant factor $b > 1$ and warm-start the optimization process from $x_t$. The key observation is that although $\mu$ decreases and $L_\mu$ increases, the other term $\|x_* - x_t\|$ is also reduced compared to $\|x_* - x_0\|$, which could cancel the blow-up effect caused by increased $L_\mu$. As a result, we expect to use the same number of iterations to optimize the smoothed function with a smaller $\mu$ such that $F(x_{2t}) - F(x_*) \le O(\mu/b)$.

To formalize our observation, we need the following key lemma.

**Lemma 1** ([21]). *For any $x \in \Omega_1$ and $\epsilon > 0$, we have*

$$\|x - x_\epsilon^\dagger\| \le \frac{dist(x_\epsilon^\dagger, \Omega_*)}{\epsilon}(F(x) - F(x_\epsilon^\dagger))$$

*where $x_\epsilon^\dagger \in S_\epsilon$ is the closest point in the $\epsilon$-sublevel set to $x$ as defined in (6).*

The lemma is proved in [21]. We include its proof in the supplement. If we apply the above bound into (9), we will see in the proof of the main theorem (Theorem 5) that the number of iterations $t$ for solving each smoothed problem is roughly $O(\frac{dist(\mathcal{L}_\epsilon, \Omega_*)}{\epsilon})$, which will be lower than $O(\frac{1}{\epsilon})$ in light of the local error bound condition given below.

**Definition 4** (Local error bound (LEB)). *A function $F(x)$ is said to satisfy a local error bound condition if there exist $\theta \in (0, 1]$ and $c > 0$ such that for any $x \in S_\epsilon$*

$$dist(x, \Omega_*) \le c(F(x) - F_*)^\theta \tag{10}$$

**Remark:** In next subsection, we will discuss the relationship with other types of conditions and show that a broad family of non-smooth functions (including almost all commonly seen functions in machine learning) obey the local error bound condition. The exponent constant $\theta$ can be considered as a local sharpness measure of the function. Figure 1 illustrates the sharpness of $F(x) = |x|^p$ for $p = 1, 1.5$, and 2 around the optimal solutions and their corresponding $\theta$.

With the local error bound condition, we can see that $dist(\mathcal{L}_\epsilon, \Omega_*) \le c\epsilon^\theta, \theta \in (0, 1]$. Now, we are ready to present the homotopy smoothing algorithm and its convergence guarantee under the

| **Algorithm 2** HOPS for solving (1) |
| --- |
| 1: **Input**: $m, t, x_0 \in \Omega_1, \epsilon_0, D^2$ and $b > 1$. |
| 2: Let $\mu_1 = \epsilon_0/(bD^2)$ |
| 3: **for** $s = 1, \ldots, m$ **do** |
| 4:     Let $x_s = \text{APG}(x_{s-1}, t, L_{\mu_s})$ |
| 5:     Update $\mu_{s+1} = \mu_s/b$ |
| 6: **end for** |
| 7: **Output**: $x_m$ |

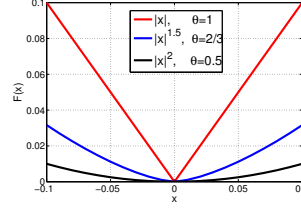

Figure 1: Illustration of local sharpness of three functions and the corresponding $\theta$ in the LEB condition.

local error bound condition. The HOPS algorithm is presented in Algorithm 2, which starts from a relatively large smoothing parameter $\mu = \mu_1$ and gradually reduces $\mu$ by a factor of $b > 1$ after running a number $t$ of iterations of APG with warm-start. The iteration complexity of HOPS is established in Theorem 5. We include the proof in the supplement.

**Theorem 5.** *Suppose Assumption 1 holds and $F(x)$ obeys the local error bound condition. Let HOPS run with $t = O(\frac{2bcD\|A\|}{\epsilon^{1-\theta}}) \geq \frac{2bcD\|A\|}{\epsilon^{1-\theta}}$ iterations for each stage, and $m = \lceil \log_b(\frac{\epsilon_0}{\epsilon}) \rceil$. Then $F(x_m) - F_* \leq 2\epsilon$. Hence, the iteration complexity for achieving an $2\epsilon$-optimal solution is $\frac{2bcD\|A\|}{\epsilon^{1-\theta}} \lceil \log_b(\frac{\epsilon_0}{\epsilon}) \rceil$ in the worst-case.*

### 4.3 Local error bounds and Applications

In this subsection, we discuss the local error bound condition and its application in non-smooth optimization problems.

**The Hoffman's bound and finding a point in a polyhedron.** A polyhedron can be expressed as $\mathcal{P} = \{x \in \mathbb{R}^d; B_1 x \leq b_1, B_2 x = b_2\}$. The Hoffman's bound [17] is expressed as

$$dist(x, \mathcal{P}) \leq c(\|(B_1 x - b_1)_+\| + \|B_2 x - b_2\|), \exists c > 0 \qquad (11)$$

where $[s]_+ = \max(0, s)$. This can be considered as the error bound for the polyhedron feasibility problem, i.e., finding a $x \in \mathcal{P}$, which is equivalent to

$$\min_{x \in \mathbb{R}^d} F(x) \triangleq \left[ \|(B_1 x - b_1)_+\| + \|B_2 x - b_2\| = \max_{u \in \Omega_2} \langle B_1 x - b_1, u_1 \rangle + \langle B_2 x - b_2, u_2 \rangle \right]$$

where $u = (u_1^\top, u_2^\top)^\top$ and $\Omega_2 = \{u | u_1 \succeq 0, \|u_1\| \leq 1, \|u_2\| \leq 1\}$. If there exists a $x \in \mathcal{P}$, then $F_* = 0$. Thus the Hoffman's bound in (11) implies a local error bound (10) with $\theta = 1$. Therefore, the HOPS has a linear convergence for finding a feasible solution in a polyhedron. If we let $\omega_+(u) = \frac{1}{2}\|u\|^2$ then $D^2 = 2$ so that the iteration complexity is $2\sqrt{2}bc \max(\|B_1\|, \|B_2\|) \lceil \log_b(\frac{\epsilon_0}{\epsilon}) \rceil$.

**Cone programming.** Let $U, V$ denote two vector spaces. Given a linear opearator $\mathcal{E} : U \to V^*$ [4], a closed convex set $\Omega \subseteq U$, and a vector $e \in V^*$, and a closed convex cone $\mathcal{K} \subseteq V$, the general constrained cone linear system (cone programing) consists of finding a vector $x \in \Omega$ such that $\mathcal{E}x - e \in \mathcal{K}^*$. Lan et al. [11] have considered Nesterov's smoothing algorithm for solving the cone programming problem with $O(1/\epsilon)$ iteration complexity. The problem can be cast into a non-smooth optimization problem:

$$\min_{x \in \Omega} F(x) \triangleq \left[ dist(\mathcal{E}x - e, \mathcal{K}^*) = \max_{\|u\| \leq 1, u \in -\mathcal{K}} \langle \mathcal{E}x - e, u \rangle \right]$$

Assume that $e \in Range(\mathcal{E}) - \mathcal{K}^*$, then $F_* = 0$. Burke et al. [5] have considered the error bound for such problems and their results imply that there exists $c > 0$ such that $dist(x, \Omega_*) \leq c(F(x) - F_*)$ as long as $\exists x \in \Omega$, s.t. $\mathcal{E}x - e \in int(\mathcal{K}^*)$, where $\Omega_*$ denotes the optimal solution set. Therefore, the HOPS also has a linear convergence for cone programming. Considering that both $U$ and $V$ are Euclidean spaces, we set $\omega_+(u) = \frac{1}{2}\|u\|^2$ then $D^2 = 1$. Thus, the iteraction complexity of HOPS for finding an $2\epsilon$-solution is $2bc\|\mathcal{E}\| \lceil \log_b(\frac{\epsilon_0}{\epsilon}) \rceil$.

**Non-smooth regularized empirical loss (REL) minimization in Machine Learning** The REL consists of a sum of loss functions on the training data and a regularizer, i.e.,

$$\min_{x \in \mathbb{R}^d} F(x) \triangleq \frac{1}{n} \sum_{i=1}^n \ell(x^\top a_i, y_i) + \lambda g(x)$$

where $(a_i, y_i), i = 1, \ldots, n$ denote pairs of a feature vector and a label of training data. Non-smooth loss functions include hinge loss $\ell(z, y) = \max(0, 1 - yz)$, absolute loss $\ell(z, y) = |z - y|$, which can be written as the max structure in (2). Non-smooth regularizers include e.g., $g(x) = \|x\|_1$, $g(x) = \|x\|_\infty$. These loss functions and regularizers are essentially piecewise linear functions, whose epigraph is a polyhedron. The error bound condition has been developed for such kind of problems [21]. In particular, if $F(x)$ has a polyhedral epigraph, then there exists $c > 0$ such that $dist(x, \Omega_*) \leq c(F(x) - F_*)$ for any $x \in \mathbb{R}^d$. It then implies HOPS has an $O(\log(\epsilon_0/\epsilon))$ iteration complexity for solving a non-smooth REL minimization with a polyhedral epigraph. Yang et al. [22] has also considered such non-smooth problems, but they only have $O(1/\epsilon)$ iteration complexity.

**When $F(x)$ is essentially locally strongly convex [9]** in terms of $\| \cdot \|$ such that [5]

$$dist^2(x, \Omega_*) \leq \frac{2}{\sigma}(F(x) - F_*), \forall x \in \mathcal{S}_\epsilon \tag{12}$$

then we can see that the local error bound holds with $\theta = 1/2$, which implies the iteration complexity of HOPS is $\tilde{O}(\frac{1}{\sqrt{\epsilon}})$, which is up to a logarithmic factor the same as the result in [6] for a strongly convex function. However, here only local strong convexity is sufficient and there is no need to develop a different algorithm and different analysis from the non-strongly convex case as done in [6]. For example, one can consider $F(x) = \|Ax - y\|_p^p = \sum_{i=1}^n |a_i^\top x - y_i|^p, p \in (1, 2)$, which satisfies (12) according to [21].

**The Kurdyka-Łojasiewicz (KL) property.** The definition of KL property is given below.

**Definition 6.** *The function $F(x)$ is said to have the KL property at $x_* \in \Omega_*$ if there exist $\eta \in (0, \infty]$, a neighborhood $U$ of $x_*$ and a continuous concave function $\varphi : [0, \eta) \to \mathbb{R}_+$ such that i) $\varphi(0) = 0$, $\varphi$ is continuous on $(0, \eta)$, ii) for all $s \in (0, \eta)$, $\varphi'(s) > 0$, iii) and for all $x \in U \cup \{x : F(x_*) < F(x) < F(x_*) + \eta\}$, the KL inequality $\varphi'(F(x) - F(x_*))\|\partial F(x)\| \geq 1$ holds.*

The function $\varphi$ is called the desingularizing function of $F$ at $x_*$, which makes the function $F(x)$ sharp by reparameterization. An important desingularizing function is in the form of $\varphi(s) = cs^{1-\beta}$ for some $c > 0$ and $\beta \in [0, 1)$, which gives the KL inequality $\|\partial F(x)\| \geq \frac{1}{c(1-\beta)}(F(x) - F(x_*))^\beta$. It has been established that the KL property is satisfied by a wide class of non-smooth functions definable in an o-minimal structure [4]. Semialgebraic functions and (globally) subanalytic functions are for instance definable in their respective classes. While the definition of KL property involves a neighborhood $U$ and a constant $\eta$, in practice many convex functions satisfy the above property with $U = \mathbb{R}^d$ and $\eta = \infty$ [1]. The proposition below shows that a function with the KL property with a desingularizing function $\varphi(s) = cs^{1-\beta}$ obeys the local error bound condition in (10) with $\theta = 1 - \beta \in (0, 1]$, which implies an iteration complexity of $\tilde{O}(1/\epsilon^\theta)$ of HOPS for optimizing such a function.

**Proposition 1.** *(Theorem 5 [10]) Let $F(x)$ be a proper, convex and lower-semicontinuous function that satisfies KL property at $x_*$ and $U$ be a neighborhood of $x_*$. For all $x \in U \cap \{x : F(x_*) < F(x) < F(x_*) + \eta\}$, if $\|\partial F(x)\| \geq \frac{1}{c(1-\beta)}(F(x) - F(x_*))^\beta$, then $dist(x, \Omega_*) \leq c(F(x) - F_*)^{1-\beta}$.*

### 4.4 Primal-Dual Homotopy Smoothing (PD-HOPS)

Finally, we note that the required number of iterations per-stage $t$ for finding an $\epsilon$ accurate solution depends on an unknown constant $c$ and sometimes $\theta$. Thus, an inappropriate setting of $t$ may lead to a less accurate solution. In practice, it can be tuned to obtain the fastest convergence. A way to eschew the tuning is to consider a primal-dual homotopy smoothing (PD-HOPS). Basically, we also apply the homotopy smoothing to the dual problem:

$$\max_{u \in \Omega_2} \Phi(u) \triangleq -\phi(u) + \min_{x \in \Omega_1} \langle A^\top u, x \rangle + g(x)$$

Denote by $\Phi_*$ the optimal value of the above problem. Under some mild conditions, it is easy to see that $\Phi_* = F_*$. By extending the analysis and result to the dual problem, we can obtain that $F(x_s) - F_* \leq \epsilon + \epsilon_s$ and $\Phi_* - \Phi(u_s) \leq \epsilon + \epsilon_s$ after the $s$-th stage with a sufficient number of iterations per-stage. As a result, we get $F(x_s) - \Phi(u_s) \leq 2(\epsilon + \epsilon_s)$. Therefore, we can use the duality gap $F(x_s) - \Phi(u_s)$ as a certificate to monitor the progress of optimization. As long as the above inequality holds, we restart the next stage. Then with at most $m = \lceil \log_b(\epsilon_0/\epsilon) \rceil$ epochs

Table 1: Comparison of different optimization algorithms by the number of iterations and running time in second (mean ± standard deviation) for achieving a solution that satisfies $F(x) - F_* \leq \epsilon$.

| | Linear Classification | | Image Denoising | | Matrix Decomposition | |
|---|---|---|---|---|---|---|
| | $\epsilon = 10^{-4}$ | $\epsilon = 10^{-5}$ | $\epsilon = 10^{-3}$ | $\epsilon = 10^{-4}$ | $\epsilon = 10^{-3}$ | $\epsilon = 10^{-4}$ |
| PD | 9861 (1.58±0.02) | 27215 (4.33±0.06) | 8078 (22.01±0.51) | 34292 (94.26±2.67) | 2523 (4.02±0.10) | 3441 (5.65±0.20) |
| APG-D | 4918 (2.44±0.22) | 28600 (11.19±0.26) | 179204 (924.37±59.67) | 1726043 (9032.69±539.01) | 1967 (6.85±0.08) | 8622 (30.36±0.11) |
| APG-F | 3277 (1.33±0.01) | 19444 (7.69±0.07) | 14150 (40.90±2.28) | 91380 (272.45±14.56) | 1115 (3.76±0.06) | 4151 (9.16±0.10) |
| HOPS-D | 1012 (0.44±0.02) | 4101 (1.67±0.01) | 3542 (13.77±0.13) | 4501 (17.38±0.10) | 224 (1.36±0.02) | 313 (1.51±0.03) |
| HOPS-F | 1009 (0.46±0.02) | 4102 (1.69±0.04) | 2206 (6.99±0.15) | 3905 (16.52±0.08) | 230 (0.91±0.01) | 312 (1.23±0.01) |
| PD-HOPS | 846 (0.36±0.01) | 3370 (1.27±0.02) | 2538 (7.97±0.13) | 3605 (11.39±0.10) | 124 (0.45±0.01) | 162 (0.64±0.01) |

we get $F(x_m) - \Phi(u_m) \leq 2(\epsilon + \epsilon_m) \leq 4\epsilon$. Similarly, we can show that PD-HOPS enjoys an $\tilde{O}(\max\{1/\epsilon^{1-\theta}, 1/\epsilon^{1-\tilde{\theta}}\})$ iteration complexity, where $\tilde{\theta}$ is the exponent constant in the local error bound of the objective function for dual problem. For example, for linear classification problems with a piecewise linear loss and $\ell_1$ norm regularizer we can have $\theta = 1$ and $\tilde{\theta} = 1$, and PD-HOPS enjoys a linear convergence. Due to the limitation of space, we defer the details of PD-HOPS and its analysis into the supplement.

## 5 Experimental Results

In this section, we present some experimental results to demonstrate the effectiveness of HOPS and PD-HOPS by comparing with two state-of-the-art algorithms, the first-order Primal-Dual (PD) method [6] and the Nesterov's smoothing with Accelerated Proximal Gradient (APG) methods. For APG, we implement two variants, where APG-D refers to the variant with the dual averaging style of update on one sequence of points (i.e., Algorithm 1) and APG-F refers to the variant of the FISTA style [2]. Similarly, we also implement the two variants for HOPS. We conduct experiments for solving three problems: (1) an $\ell_1$-norm regularized hinge loss for linear classification on the w1a dataset [6]; (2) a total variation based ROF model for image denoising on the Cameraman picture [7]; (3) a nuclear norm regularized absolute error minimization for low-rank and sparse matrix decomposition on a synthetic data. More details about the formulations and experimental setup can be found in the supplement.

To make fair comparison, we stop each algorithm when the optimality gap is less than a given $\epsilon$ and count the number of iterations and the running time that each algorithm requires. The optimal value is obtained by running PD with a sufficiently large number of iterations such that the duality gap is very small. We present the comparison of different algorithms on different tasks in Table 1, where for PD-HOPS we only report the results of using the faster variant of APG, i.e., APG-F. We repeat each algorithm 10 times for solving a particular problem and then report the averaged running time in second and the corresponding standard deviations. The running time of PD-HOPS only accounts the time for updating the primal variable since the updates for the dual variable are fully decoupled from the primal updates and can be carried out in parallel. From the results, we can see that (i) HOPS converges consistently faster than their APG variants especially when $\epsilon$ is small; (ii) PD-HOPS allows for choosing the number of iterations at each epoch automatically, yielding faster convergence speed than HOPS with manual tuning; (iii) both HOPS and PD-HOPS are significantly faster than PD.

## 6 Conclusions

In this paper, we have developed a homotopy smoothing (HOPS) algorithm for solving a family of structured non-smooth optimization problems with formal guarantee on the iteration complexities. We show that the proposed HOPS can achieve a lower iteration complexity of $\tilde{O}(1/\epsilon^{1-\theta})$ with $\theta \in (0, 1]$ for obtaining an $\epsilon$-optimal solution under a mild local error bound condition. The experimental results on three different tasks demonstrate the effectiveness of HOPS.

## Acknowlegements

We thank the anonymous reviewers for their helpful comments. Y. Xu and T. Yang are partially supported by National Science Foundation (IIS-1463988, IIS-1545995).

## Footnotes

[2]The algorithm in [16] was developed for handling a smooth component $g(x)$, which can be extended to handling a non-smooth component $g(x)$ whose proximal mapping is easy to compute.

[3] If $g(x)$ is smooth, this assumption can be relaxed. We will defer the discussion and result on a smooth function $g(x)$ to the supplement.

[4] $V^*$ represents the dual space of $V$. The notations and descriptions are adopted from [11].

[5]This is true if $g(x)$ is strongly convex or locally strongly convex.

[6] https://www.csie.ntu.edu.tw/~cjlin/libsvmtools/datasets/

[7] http://pages.cs.wisc.edu/~swright/TVdenoising/

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
