[Supplementary Material]

# Supplement of "Homotopy Smoothing for Non-Smooth Problems with Lower Complexity than $O(1/\epsilon)$"

**Yi Xu[†], Yan Yan[‡], Qihang Lin[♮], Tianbao Yang[†]**
[†] Department of Computer Science, University of Iowa, Iowa City, IA 52242
[‡] QCIS, University of Technology Sydney, NSW 2007, Australia
[♮] Department of Management Sciences, University of Iowa, Iowa City, IA 52242
{yi-xu, qihang-lin, tianbao-yang}@uiowa.edu, yan.yan-3@student.uts.edu.au

## 1 Proofs

**Lemma 1** ([2]). *For any $x \in \Omega_1$ and $\epsilon > 0$, we have*

$$\|x - x_\epsilon^\dagger\| \leq \frac{dist(x_\epsilon^\dagger, \Omega_*)}{\epsilon}(F(x) - F(x_\epsilon^\dagger))$$

*where $x_\epsilon^\dagger \in \mathcal{S}_\epsilon$ is the closest point in the $\epsilon$-sublevel set to $x$.*

The lemma is an immediate result from [2]. For completeness, we give the proof here.

### 1.1 Proof of Lemma 1

*Proof.* Consider $\|x\|$ to be an Euclidean norm. We first recall the definition of $x_\epsilon^\dagger$:

$$x_\epsilon^\dagger = \arg\min_{z \in \mathcal{S}_\epsilon} \|z - x\|^2 \tag{1}$$

where $\mathcal{S}_\epsilon = \{x \in \Omega_1 : F(x) \leq F_* + \epsilon\}$ is the sublevel set. We assume $x \notin \mathcal{S}_\epsilon$, otherwise the conclusion holds trivially. Thus $F(x_\epsilon^\dagger) = F_* + \epsilon$. By the first-order optimality conditions of (1), we have for any $z \in \Omega_1$, there exists $\zeta \geq 0$ (the Lagrangian multiplier of problem (1))

$$(x_\epsilon^\dagger - x + \zeta\partial F(x_\epsilon^\dagger))^\top(z - x_\epsilon^\dagger) \geq 0 \tag{2}$$

Let $z = x$ we have

$$\zeta\partial F(x_\epsilon^\dagger)^\top(x - x_\epsilon^\dagger) \geq \|x - x_\epsilon^\dagger\|^2$$

We argue that $\zeta > 0$, otherwise $x = x_\epsilon^\dagger$ contradicting to the assumption $x \notin \mathcal{S}_\epsilon$. Therefore

$$F(x) - F(x_\epsilon^\dagger) \geq \partial F(x_\epsilon^\dagger)^\top(x - x_\epsilon^\dagger) \geq \frac{\|x - x_\epsilon^\dagger\|^2}{\zeta} = \frac{\|x - x_\epsilon^\dagger\|}{\zeta}\|x - x_\epsilon^\dagger\| \tag{3}$$

Next we prove that $\zeta$ is upper bounded. Since

$$-\epsilon = F(x_\epsilon^*) - F(x_\epsilon^\dagger) \geq (x_\epsilon^* - x_\epsilon^\dagger)^\top \partial F(x_\epsilon^\dagger)$$

where $x_\epsilon^*$ is the closest point to $x_\epsilon^\dagger$ in the optimal set. Let $z = x_\epsilon^*$ in the inequality of (2), we have

$$(x_\epsilon^\dagger - x)^\top(x_\epsilon^* - x_\epsilon^\dagger) \geq \zeta(x_\epsilon^\dagger - x_\epsilon^*)^\top \partial F(x_\epsilon^\dagger) \geq \zeta\epsilon$$

Thus

$$\zeta \leq \frac{(x_\epsilon^\dagger - x)^\top(x_\epsilon^* - x_\epsilon^\dagger)}{\epsilon} \leq \frac{dist(x_\epsilon^\dagger, \Omega_*)\|x_\epsilon^\dagger - x\|}{\epsilon}$$

Therefore

$$\frac{\|x - x_\epsilon^\dagger\|}{\zeta} \geq \frac{\epsilon}{dist(x_\epsilon^\dagger, \Omega_*)}$$

Combining the above inequality with (3) we have

$$\|x - x_\epsilon^\dagger\| \leq \frac{dist(x_\epsilon^\dagger, \Omega_*)}{\epsilon}(F(x) - F(x_\epsilon^\dagger))$$

which completes the proof.

□

## 1.2  Proof of Theorem 5

*Proof.* Let $x_{s,\epsilon}^\dagger$ denote the closest point to $x_s$ in the $\epsilon$ sublevel set. Define $\epsilon_s \triangleq \frac{\epsilon_0}{b^s}$. Note that $\mu_s = \epsilon_s/D^2$. We will show by induction that $F(x_s) - F_* \leq \epsilon_s + \epsilon$ for $s = 0, 1, \ldots$ which leads to our conclusion when $s = m$. The inequality holds obviously for $s = 0$. Assuming $F(x_{s-1}) - F_* \leq \epsilon_{s-1} + \epsilon$, we need to show that $F(x_s) - F_* \leq \epsilon_s + \epsilon$. We apply Corollary 3 to the $s$-th epoch of Algorithm 2 and get

$$F(x_s) - F(x_{s-1,\epsilon}^\dagger) \leq \frac{D^2 \mu_s}{2} + \frac{2\|A\|^2 \|x_{s-1} - x_{s-1,\epsilon}^\dagger\|^2}{\mu_s t^2} \tag{4}$$

First, we assume $F(x_{s-1}) - F_* \leq \epsilon$, i.e. $x_{s-1} \in \mathcal{S}_\epsilon$. Then we have $x_{s-1,\epsilon}^\dagger = x_{s-1}$ and

$$F(x_s) - F(x_{s-1,\epsilon}^\dagger) \leq \frac{D^2 \mu_s}{2} \leq \frac{\epsilon_s}{2}$$

As a result,

$$F(x_s) - F_* \leq F(x_{s-1,\epsilon}^\dagger) - F_* + \frac{\epsilon_s}{2} \leq \epsilon + \epsilon_s$$

Next, we consider $F(x_{s-1}) - F_* > \epsilon$, i.e. $x_{s-1} \notin \mathcal{S}_\epsilon$. Then we have $F(x_{s-1,\epsilon}^\dagger) - F_* = \epsilon$. By Lemma 1, we have

$$\begin{aligned}
\|x_{s-1} - x_{s-1,\epsilon}^\dagger\| &\leq \frac{dist(x_{s-1,\epsilon}^\dagger, \Omega_*)}{\epsilon}(F(x_{s-1}) - F(x_{s-1,\epsilon}^\dagger)) \\
&\leq \frac{dist(x_{s-1,\epsilon}^\dagger, \Omega_*)}{\epsilon}[\epsilon_{s-1} + \epsilon - \epsilon] = \frac{dist(x_{s-1,\epsilon}^\dagger, \Omega_*)\epsilon_{s-1}}{\epsilon} \\
&\leq \frac{c(F(x_{s-1,\epsilon}^\dagger) - F_*)^\theta \epsilon_{s-1}}{\epsilon} \\
&\leq \frac{c(\epsilon)^\theta \epsilon_{s-1}}{\epsilon} = \frac{c\epsilon_{s-1}}{\epsilon^{1-\theta}}
\end{aligned} \tag{5}$$

Combining (4) and (5) and using the fact that $\mu_s = \frac{\epsilon_s}{D^2}$ and $t \geq \frac{2bcD\|A\|}{\epsilon^{1-\theta}}$, we have

$$F(x_s) - F(x_{s-1,\epsilon}^\dagger) \leq \frac{\epsilon_s}{2} + \frac{\epsilon_{s-1}^2}{2\epsilon_s b^2} = \epsilon_s$$

which together with the fact that $F(x_{s-1,\epsilon}^\dagger) = F_* + \epsilon$ implies

$$F(x_s) - F_* \leq \epsilon + \epsilon_s$$

Therefore by induction, we have

$$F(x_m) - F_* \leq \epsilon_m + \epsilon = \frac{\epsilon_0}{b^m} + \epsilon \leq 2\epsilon$$

where the last inequality is due to the value of $m$.

□

**Algorithm 3** An Accelerated Proximal Gradient Method ($g$ is smooth): APG$(x_0, t, L_\mu)$

1: **Input**: the number of iterations $t$, the initial solution $x_0$, and the smoothness constant $L_\mu$
2: Let $\theta_0 = 1$, $U_{-1} = 0$, $z_0 = x_0$
3: Let $\alpha_k$ and $\theta_k$ be two sequences given in Theorem 2.
4: **for** $k = 0, \ldots, t-1$ **do**
5:    Compute $y_k = (1 - \theta_k)x_k + \theta_k z_k$
6:    Compute $u_k = \nabla f_\mu(y_k) + \nabla g(y_k)$, $U_k = U_{k-1} + \frac{u_k}{\alpha_k}$
7:    Compute $z_{k+1} = \widetilde{\Pi}_{U_k}^{(L_\mu + M)/\sigma_1}(x_0)$ and $x_{k+1} = \widetilde{\Pi}_{u_k}^{L_\mu + M}(y_k)$
8: **end for**
9: **Output**: $x_t$

## 2   HOPS with a smooth $g(x)$

In the Preliminaries section, we assume that $g(z)$ is simple enough such that the proximal mapping defined below is easy to compute:

$$P_{\lambda g}(x) = \min_{z \in \Omega_1} \frac{1}{2}\|z - x\|^2 + \lambda g(z) \tag{6}$$

We claimed that if $g(z)$ is smooth, this assumption can be relaxed. In this section, we present the discussion and result for a smooth function $g(x)$ without assuming that its proximal mapping is easy to compute. In particular, we will consider $g$ as a smooth component in $f_\mu + g$ and use the gradient of both $f_\mu$ and $g$ in the updating. The detailed updates are presented in Algorithm 3, where

$$\widetilde{\Pi}_u^c(x) = \arg\min_{z \in \Omega_1} \langle u, z \rangle + \frac{c}{2}\|z - x\|^2 \tag{7}$$

To present the convergence guarantee, we assume that the function $g$ is $M$-smooth w.r.t $\|x\|$, then the smoothness parameter of objective function $F_\mu(x) = f_\mu(x) + g(x)$ is

$$L = L_\mu + M = \frac{\|A\|^2}{\mu} + M \tag{8}$$

Then, we state the convergence result of Algorithm 3 in the following corollary.

**Corollary 6.** Let $\theta_k = \frac{2}{k+2}$, $\alpha_k = \frac{2}{k+1}$, $k \geq 0$ or $\alpha_{k+1} = \theta_{k+1} = \frac{\sqrt{\theta_k^4 + 4\theta_k^2} - \theta_k^2}{2}$, $k \geq 0$. For any $x \in \Omega_1$, we have

$$F(x_t) - F(x) \leq \frac{\mu D^2}{2} + \frac{2\|A\|^2\|x - x_0\|^2}{\mu t^2} + \frac{2M\|x - x_0\|^2}{t^2} \tag{9}$$

**Remark:** In order to have $F(x_t) \leq F(x_*) + \epsilon$, we can consider $x = x_*$ in Corollary 6, i.e.

$$F(x_t) - F(x_*) \leq \frac{\mu D^2}{2} + \frac{2\|A\|^2\|x_* - x_0\|^2}{\mu t^2} + \frac{2M\|x_* - x_0\|^2}{t^2} \tag{10}$$

In particular, we set

$$\mu = \frac{2\epsilon}{3D^2} \tag{11}$$

and

$$t \geq \max\left\{ \frac{3D\|A\|\|x_* - x_0\|}{\epsilon}, \frac{\sqrt{6M}\|x_* - x_0\|}{\sqrt{\epsilon}} \right\} \tag{12}$$

Algorithm 3 also achieves the iteration complecity of $O(1/\epsilon)$.

Similarly, we can develop the HOPS algorithm and present it in Algorithm 4. The iteration complexity of HOPS is established in Theorem 7.

---
**Algorithm 4** Homotopy Smoothing (HOPS) for solving (1) ($g$ is smooth)
---
1: **Input**: the number of stages $m$ and the number of iterations $t$ per-stage, and the initial solution $x_0 \in \Omega_1$ and a parameter $b > 1$.
2: Let $\mu_1 = \frac{2\epsilon_0}{3bD^2}$
3: **for** $s = 1, \ldots, m$ **do**
4:     Let $x_s = \text{APG}(x_{s-1}, t, L_{\mu_s})$
5:     Update $\mu_{s+1} = \mu_s / b$
6: **end for**
7: **Output**: $x_m$
---

**Theorem 7.** *Suppose Assumption 1 holds and $F(x)$ obeys the local error bound condition. Let HOPS run with $t = O(1/\epsilon^{1-\theta}) \geq \max \left\{ \frac{3D\|A\|bc}{\epsilon^{1-\theta}}, \frac{\sqrt{6M\epsilon_s}bc}{\epsilon^{1-\theta}} \right\}$ iterations for each stage, and $m = \lceil \log_b(\frac{\epsilon_0}{\epsilon}) \rceil$. Then*

$$F(x_m) - F_* \leq 2\epsilon.$$

*Hence, the iteration complexity for achieving an $2\epsilon$-optimal solution is $\widetilde{O}(1/\epsilon^{1-\theta})$.*

*Proof.* Let $x_{s,\epsilon}^\dagger$ denote the closest point to $x_s$ in the $\epsilon$ sublevel set and define $\epsilon_s \triangleq \frac{\epsilon_0}{b^s}$. We will show by induction that $F(x_s) - F_* \leq \epsilon_s + \epsilon$ for $s = 0, 1, \ldots$ which leads to our conclusion when $s = m$. The inequality holds obviously for $s = 0$. Assuming $F(x_{s-1}) - F_* \leq \epsilon_{s-1} + \epsilon$, we need to show that $F(x_s) - F_* \leq \epsilon_s + \epsilon$. We apply Corollary 6 to the $s$-th epoch of Algorithm 3 and get

$$F(x_s) - F(x_{s-1,\epsilon}^\dagger) \leq \frac{\mu_s D^2}{2} + \frac{2\|A\|^2\|x_{s-1,\epsilon}^\dagger - x_{s-1}\|^2}{\mu_s t^2} + \frac{2M\|x_{s-1,\epsilon}^\dagger - x_{s-1}\|^2}{t^2} \tag{13}$$

First, we assume $F(x_{s-1}) - F_* \leq \epsilon$, i.e. $x_{s-1} \in \mathcal{S}_\epsilon$. Then we have $x_{s-1,\epsilon}^\dagger = x_{s-1}$ and

$$F(x_s) - F(x_{s-1,\epsilon}^\dagger) \leq \frac{D^2 \mu_s}{2} \leq \frac{\epsilon_s}{3}$$

As a result,

$$F(x_s) - F_* \leq F(x_{s-1,\epsilon}^\dagger) - F_* + \frac{\epsilon_s}{3} \leq \epsilon + \epsilon_s$$

Next, we consider $F(x_{s-1}) - F_* > \epsilon$, i.e. $x_{s-1} \notin \mathcal{S}_\epsilon$. Then we have $F(x_{s-1,\epsilon}^\dagger) - F_* = \epsilon$. Recall that

$$\|x_{s-1} - x_{s-1,\epsilon}^\dagger\| \leq \frac{c\epsilon_{s-1}}{\epsilon^{1-\theta}} \tag{14}$$

Combining (13) and (14) and using the fact that $\mu_s = \frac{2\epsilon_s}{3D^2}$ and $t \geq \max \left\{ \frac{3D\|A\|bc}{\epsilon^{1-\theta}}, \frac{\sqrt{6M\epsilon_s}bc}{\epsilon^{1-\theta}} \right\}$, we get

$$F(x_s) - F(x_{s-1,\epsilon}^\dagger) \leq \frac{\epsilon_s}{3} + \frac{3D^2\|A\|^2 c^2 \epsilon_{s-1}^2}{\epsilon_s \epsilon^{2(1-\theta)} t^2} + \frac{2Mc^2 \epsilon_{s-1}^2}{\epsilon^{2(1-\theta)} t^2}$$

$$\leq \frac{\epsilon_s}{3} + \frac{\epsilon_{s-1}^2}{3\epsilon_s b^2} + \frac{\epsilon_{s-1}^2}{3\epsilon_s b^2} = \epsilon_s$$

which together with the fact that $F(x_{s-1,\epsilon}^\dagger) = F_* + \epsilon$ implies

$$F(x_s) - F_* \leq \epsilon + \epsilon_s$$

Therefore by induction, we have

$$F(x_m) - F_* \leq \epsilon_m + \epsilon = \frac{\epsilon_0}{b^m} + \epsilon \leq 2\epsilon$$

where the last inequality is due to the value of $m = \lceil \log_b(\frac{\epsilon_0}{\epsilon}) \rceil$.

In fact, the number of iteration in each stage depends on $s$, then the iteration complexity for achieving an $2\epsilon$-optimal solution is

$$\sum_{s=1}^{m} \max\left\{ \frac{3D\|A\|bc}{\epsilon^{1-\theta}}, \frac{\sqrt{6M\epsilon_s}bc}{\epsilon^{1-\theta}} \right\} \leq \sum_{s=1}^{m} \frac{3D\|A\|bc + \sqrt{6M\epsilon_s}bc}{\epsilon^{1-\theta}}$$

$$= \frac{3D\|A\|bc}{\epsilon^{1-\theta}} \left\lceil \log_b\left(\frac{\epsilon_0}{\epsilon}\right)\right\rceil + \sum_{s=1}^{m} \frac{\sqrt{6M\epsilon_0}bc}{\sqrt{b^s}\epsilon^{1-\theta}}$$

$$\leq \frac{3D\|A\|bc}{\epsilon^{1-\theta}} \left\lceil \log_b\left(\frac{\epsilon_0}{\epsilon}\right)\right\rceil + \frac{\sqrt{6M\epsilon_0}bc}{(\sqrt{b}-1)\epsilon^{1-\theta}}$$

$\square$

## 3 Primal-Dual Homotopy Smoothing

We note that the required number of iterations per-stage $t$ for finding an $\epsilon$ accurate solution depends on unknown constant $c$ and sometimes $\theta$. Thus, an inappropriate setting of $t$ may lead to a less accurate solution. To address this issue, we present a primal-dual homotopy smoothing. Basically, we also apply the homotopy smoothing to the dual problem:

$$\max_{u \in \Omega_2} \Phi(u) \triangleq -\phi(u) + \underbrace{\min_{x \in \Omega_1} \langle A^\top u, x\rangle + g(x)}_{\psi(u)} \tag{15}$$

Denote by $\Phi_*$ the optimal value of the above problem. It is easy to see that $\Phi_* = F_*$. By extending the analysis and result to the dual problem, we can obtain that $F(x_m) - \Phi(u_m) \leq 4\epsilon$. Thus, we can use the duality gap $F(x_s) - \Phi(u_s)$ as a certificate to monitor the progress of optimization. In this section, we present more details.

### 3.1 Nesterov's smoothing on the Dual problem

We construct a smooth function from $\psi_\eta(u)$ that well approximates $\psi(u)$:

$$\psi_\eta(u) = \min_{x \in \Omega_1} \langle A^\top u, x\rangle + g(x) + \eta\omega(x)$$

where $\omega(x)$ is a 1-strongly convex function w.r.t. $x$ in terms of a norm $\|\cdot\|$ [1]. Similarly, we know that $\psi_\eta(u)$ is a smooth function of $u$ with respect to an Euclidean norm $\|u\|$ with smoothness parameter $L_\eta = \frac{1}{\eta}\|A\|_+^2$, where $\|A\|_+$ is defined by $\|A\|_+ = \max_{\|x\|\leq 1} \max_{\|u\|_+\leq 1}\langle A^\top u, x\rangle$. Denote by

$$x_\eta(u) = \arg\min_{x \in \Omega_1} \langle A^\top u, x\rangle + g(x) + \eta\omega(x)$$

The gradient of $\psi_\eta(u)$ is computed by $\nabla\psi_\eta(u) = Ax_\eta(u)$. We can see that when $\eta$ is very small, $\psi_\eta(u)$ gives a good approximation of $\psi(u)$. This motivates us to solve the following composite optimization problem

$$\max_{u \in \Omega_2} \Phi_\eta(u) \triangleq -\phi(u) + \psi_\eta(u)$$

Similar to solving the primal problem, an accelerated proximal gradient method for dual problem can be employed to solve the above problem. We present the details in Algorithm 5. We present the convergence results for Algorithm 5 in the following theorem:

**Theorem 8.** *Let* $\theta_k = \frac{2}{k+2}$, $\alpha_k = \frac{2}{k+1}$, $k \geq 0$ *or* $\alpha_{k+1} = \theta_{k+1} = \frac{\sqrt{\theta_k^4 + 4\theta_k^2} - \theta_k^2}{2}$, $k \geq 0$. *For any* $u \in \Omega_2$, *we have*

$$\Phi_\eta(u) - \Phi_\eta(u_t) \leq \frac{2L_\eta\|u - u_0\|^2}{t^2} \tag{16}$$

---

**Algorithm 5** An Accelerated Proximal Gradient Method for the dual problem: DAPG$(u_0, t, L_\eta)$

---

1: **Input**: the number of iterations $t$, the initial solution $u_0$, and the smoothness constant $L_\eta$
2: Let $\theta_0 = 1$, $V_{-1} = 0$, $\Gamma_{-1} = 0$, $r_0 = u_0$
3: Let $\alpha_k$ and $\theta_k$ be two sequences given in Theorem 8.
4: **for** $k = 0, \ldots, t - 1$ **do**
5:     Compute $w_k = (1 - \theta_k)u_k + \theta_k r_k$
6:     Compute $v_k = \nabla \psi_\eta(w_k)$, $V_k = V_{k-1} - \frac{v_k}{\alpha_k}$, and $\Gamma_k = \Gamma_{k-1} + \frac{1}{\alpha_k}$
7:     Compute $r_{k+1} = \Pi_{V_k, \Gamma_k \phi}^{L_\eta / \sigma_2}(u_0)$ and $u_{k+1} = \Pi_{-v_k, \phi}^{L_\eta}(w_k)$
8: **end for**
9: **Output**: $u_t$

---

---

**Algorithm 6** Homotopy Smoothing (HOPS) for solving dual problem

---

1: **Input**: the number of stages $m$ and the number of iterations $t$ per-stage, and the initial solution $u_0 \in \Omega_2$ and a parameter $b > 1$.
2: Let $\eta_1 = \epsilon_0 / (b\tilde{D}^2)$
3: **for** $s = 1, \ldots, m$ **do**
4:     Let $u_s = \text{DAPG}(u_{s-1}, t, L_{\eta_s})$
5:     Update $\eta_{s+1} = \eta_s / b$
6: **end for**
7: **Output**: $u_m$

---

### 3.2 HOPS for the Dual Problem

Similar to primal problem, we can also develop the HOPS for dual problem, which is presented in Algorithm 6. A convergence can be established similarly by exploring a local error bound condition on $\Phi(u)$. To present the convergence result, we make the following assumptions, which are similar as the primal problem.

**Assumption 9.** *For a concave maximization problem (15), we assume (i) there exist $u_0 \in \Omega_2$ and $\epsilon_0 \geq 0$ such that $\max_{u \in \Omega_2} \Phi(u) - \Phi(u_0) \leq \epsilon_0$; (ii) let $\psi(u) = \min_{x \in \Omega_1} \langle A^\top u, x \rangle + g(x)$, where $g(x)$ is a convex function; (iii) There exists a constant $\tilde{D}$ such that $\max_{x \in \Omega_1} \omega(x) \leq \tilde{D}^2 / 2$.*

Let $\widetilde{\Omega}_*$ denote the optimal solution set of (15). For any $u \in \Omega_2$, let $u^*$ denote the closest optimal solution in $\widetilde{\Omega}_*$ to $u$, i.e., $u^* = \arg\min_{v \in \widetilde{\Omega}_*} \|v - u\|^2$. We denote by $\widetilde{\mathcal{L}}_\epsilon$ the $\epsilon$-level set of $\Phi(u)$ and by $\widetilde{\mathcal{S}}_\epsilon$ the $\epsilon$-sublevel set of $\Phi(u)$, respectively, i.e.,

$$\widetilde{\mathcal{L}}_\epsilon = \{u \in \Omega_2 : \Phi(u) = \Phi_* - \epsilon\}, \quad \widetilde{\mathcal{S}}_\epsilon = \{u \in \Omega_2 : \Phi(u) \geq \Phi_* - \epsilon\} \tag{17}$$

A local error bound condition is also imposed.

**Definition 10** (Local error bound). *A function $\Phi(u)$ is said to satisfy a local error bound condition if there exist $\tilde{\theta} \in (0, 1]$ and $\tilde{c} > 0$ such that for any $u \in \widetilde{\mathcal{S}}_\epsilon$*

$$dist(u, \widetilde{\Omega}_*) \leq \tilde{c}(\Phi_* - \Phi(u))^{\tilde{\theta}} \tag{18}$$

**Theorem 11.** *Suppose Assumption 9 holds and $\Phi(u)$ obeys the local error bound condition. Let HOPS for dual problem run with $t = O\left(\frac{2b\tilde{c}\tilde{D}\|A\|_+}{\epsilon^{1-\tilde{\theta}}}\right) \geq \frac{2b\tilde{c}\tilde{D}\|A\|_+}{\epsilon^{1-\tilde{\theta}}}$ iterations for each stage, and $m = \lceil \log_b(\frac{\epsilon_0}{\epsilon}) \rceil$. Then*

$$\Phi_* - \Phi(u_m) \leq 2\epsilon.$$

*Hence, the iteration complexity for achieving an $2\epsilon$-optimal solution is $\frac{2b\tilde{c}\tilde{D}\|A\|_+}{\epsilon^{1-\tilde{\theta}}} \lceil \log_b(\frac{\epsilon_0}{\epsilon}) \rceil$ in the worst-case.*

The above theorem can be proved similarly as Theorem 5.

**Algorithm 7** Primal-Dual Homotopy Smoothing (PD-HOPS) for solving (1)

1: **Input**: the number of stages $m$, initial solutions $x_0 \in \Omega_1, u_0 \in \Omega_2$ and a parameter $b > 1$.
2: Let $\epsilon_1 = \frac{\epsilon_0}{b}, \mu_1 = \frac{\epsilon_1}{D^2}, \eta_1 = \frac{\epsilon_1}{D^2}$
3: **for** $s = 1, \ldots, m$ **do**
4:    **for** $k = 0, 1, \ldots,$ **do**
5:       Update the sequence of $x_{k+1}$ as in Algorithm 1 starting from $x_{s-1}$
6:       Update the sequence of $u_{k+1}$ as in Algorithm 5 starting from $u_{s-1}$
7:       Check occasionally if $F(x_{k+1}) - \Phi(u_{k+1}) \leq 2(\epsilon_s + \epsilon)$; break the loop if it is true
8:    **end for**
9:    Update $x_s = x_{k+1}$ and $u_s = u_{k+1}$
10:    Update $\epsilon_{s+1} = \epsilon_s/b, \mu_{s+1} = \mu_s/b$ and $\eta_{s+1} = \eta_s/b$
11: **end for**
12: **Output**: $(x_m, u_m)$

### 3.3 Primal-Dual HOPS

As mentioned before, we can use the duality gap $F(x_s) - \Phi(u_s)$ as a certificate to monitor the progress of optimization to address the problem of detecting the number of iterations per-stage $t$. We describe the details in Algorithm 7. Following the analysis as in the proof of Theorem 5, when the number of iterations in the $s$-th epoch denoted by $t_s$ satisfies $t_s \geq \max\{\frac{2bcD\|A\|}{\epsilon^{1-\theta}}, \frac{2b\tilde{c}\tilde{D}\|A\|_+}{\epsilon^{1-\tilde{\theta}}}\}$, we can have $F(x_s) - F_* \leq \epsilon + \epsilon_s$ and $\Phi_* - \Phi(u_s) \leq \epsilon + \epsilon_s$, so that

$$F(x_s) - \Phi(u_s) \leq 2(\epsilon + \epsilon_s) \tag{19}$$

Hence, as long as the above condition satisfies, we restart the next stage. Then with at most $m = \lceil \log_b(\epsilon_0/\epsilon) \rceil$ epochs we have

$$F(x_m) - \Phi(u_m) \leq 2(\epsilon + \epsilon_m) \leq 4\epsilon. \tag{20}$$

## 4 Experimental Design

We conduct experiments for solving three problems: (1) an $\ell_1$-norm regularized hinge loss for linear classification on the w1a dataset; (2) a total variation based ROF model for image denoising on the Cameraman picture; (3) a nuclear norm regularized absolute error minimization for low-rank and sparse matrix decomposition on a synthetic data. The three problems are discussed in details below.

- **Linear Classification:** In linear classification problems, the goal is to solve the following optimization problem:

$$\min_{x \in \mathbb{R}^d} \quad \frac{1}{n} \sum_{i=1}^{n} \ell(x^\top a_i, y_i) + \lambda r(x)$$

  where $(a_i, y_i), i = 1, 2, \ldots, n$ denote pairs of and label of training data, $\ell(x^\top a_i, y_i)$ is loss function, $r(x)$ is regularizer, and $\lambda$ is regularization parameter. In our experiment, we use the hinge loss (a non-smooth function) $\ell(zy) = \max(0, 1 - zy) = \max_{\alpha \in [0,1]} \alpha(1 - zy)$ for loss function and the $\ell_1$-norm for regularizer:

$$\min_{x \in \mathbb{R}^d} F(x) \triangleq \frac{1}{n} \sum_{i=1}^{n} \max_{u_i \in [0,1]} u_i(1 - y_i a_i^\top x) + \lambda\|x\|_1 \tag{21}$$

  We first write (21) into the following equivalent minimax formulation

$$\min_{x \in \mathbb{R}^d} \max_{u \in [0,1]^n} u^\top A x + \frac{u^\top \mathbf{1}}{n} + \lambda\|x\|_1 \tag{22}$$

where matrix $A = -\frac{1}{n}(y_1 a_1, y_2 a_2, \ldots, y_n a_n)^\top$ and $\mathbf{1}$ is a vector of all ones. Thus, $f(x) = \max_{u \in [0,1]^n} u^\top A x + \frac{u^\top \mathbf{1}}{n}$ and $g(x) = \lambda \|x\|_1$. To apply Nesterov's smoothing technique, we construct the following smoothed function

$$f_\mu(x) = \max_{u \in [0,1]^n} u^\top A x + \frac{u^\top \mathbf{1}}{n} - \frac{\mu}{2}\|u\|_2^2 \qquad (23)$$

We construct the experiment on the w1a dataset, which contains $2,477$ training examples and $300$ features. We fix the regularization parameter $\lambda = n^{-1}$.

- **Image Denoising:** For total variation (TV) based image denoising problem, we consider the following ROF model:

$$\min_x \int_\Omega |\nabla x| + \frac{\lambda}{2}\|x - h\|_2^2, \qquad (24)$$

where $h$ is the observed noisy image, $\Omega \subset \mathbb{R}^{m \times n}$ is the image domain, $\int_\Omega |\nabla x|$ is the TV regularization term, and $\lambda$ is the trade-off parameter between regularization and fidelity. Following the ROF setting in [1], we obtain the following discrete version:

$$\min_{x \in X} F(x) \triangleq \|\nabla x\|_1 + \frac{\lambda}{2}\|x - h\|_2^2. \qquad (25)$$

where $X = \mathbb{R}^{mn}$ is a finite dimensional vector space, $\nabla x \in Y$ and $Y = X \times X$. The discrete gradient operator $\nabla x$ is defined as following that has two components:

$$(\nabla x)_{i,j}^1 = \begin{cases} x_{i+1,j} - x_{i,j} & \text{if } i < m \\ 0 & \text{if } i = m \end{cases}$$

$$(\nabla x)_{i,j}^2 = \begin{cases} x_{i,j+1} - x_{i,j} & \text{if } j < n \\ 0 & \text{if } j = n, \end{cases}$$

and $\|\nabla x\|_1$ is defined as

$$\|\nabla x\|_1 = \sum_{i,j} |(\nabla x)_{i,j}| = \sum_{i,j} \sqrt{((\nabla x)_{i,j}^1)^2 + ((\nabla x)_{i,j}^2)^2}.$$

According to [1], we have the minimax formulation of ROF model as

$$\min_{x \in X} \max_{u \in \Omega_2} -\langle x, \text{div} u \rangle + \frac{\lambda}{2}\|x - h\|_2^2 \qquad (26)$$

where $\Omega_2 = \{u : u \in Y, \|u\|_\infty \leq 1\}$, $\|u\|_\infty = \max_{i,j} \sqrt{(u_{i,j}^1)^2 + (u_{i,j}^2)^2}$, and $\text{div} u$ is the discrete divergence operator [1]. Thus, $f(x) = \max_{u \in \Omega_2} -\langle x, \text{div} u \rangle$ and $g(x) = \frac{\lambda}{2}\|x - h\|_2^2$. By using Nesterov's smoothing technique, we have the following smoothed function

$$\max_{u \in \Omega_2} -\langle x, \text{div} u \rangle - \frac{\mu}{2}\|u\|_2^2. \qquad (27)$$

In our experiment, we use Cameraman picture of size $256 \times 256$ with additive zero mean Gaussian noise with standard deviation $\sigma = 0.05$ and we set $\lambda = 20$.

- **Matrix Decomposition:** In low-rank and sparse matrix decomposition problem, suppose given a data matrix $O \in \mathbb{R}^{m \times n}$, we aim to decompose it as

$$O = X + E$$

where $X \in \mathbb{R}^{m \times n}$ is a low-rank matrix, and $E \in \mathbb{R}^{m \times n}$ represents errors and it is sparse. We use nuclear norm regularized absolute error minimization:

$$\min_{X \in \mathbb{R}^{m \times n}} F(X) = \|X\|_* + \lambda\|E\|_1$$
$$\text{s.t. } O = X + E$$

where $\|X\|_* = \sum_i \sigma_i(X)$ denotes the nuclear norm of matrix $X$, i.e., the summation of singular values of matrix $X$, and $\|E\|_1 = \sum_{ij} |E_{ij}|$ denotes the $\ell_1$-norm of $E$. The above formulation is equavilent to

$$\min_{X \in \mathbb{R}^{m \times n}} F(X) = \|X\|_* + \lambda \|O - X\|_1 \qquad (28)$$

We first write (28) into the following equivalent minimax formulation

$$\min_{X \in \mathbb{R}^{m \times n}} \max_{\|U\|_\infty \leq 1} -\lambda \langle X, U \rangle + \lambda \langle O, U \rangle + \|X\|_* \qquad (29)$$

where $U \in \mathbb{R}^{m \times n}$ and $\|U\|_\infty = \max_{ij} |U_{ij}|$. Thus, $f(X) = \max_{\|U\|_\infty \leq 1} -\lambda \langle X, U \rangle + \lambda \langle O, U \rangle$ and $g(X) = \|X\|_*$. To apply Nesterov's smoothing technique, we consider the following smoothed function

$$f_\mu(X) = \max_{\|U\|_\infty \leq 1} -\lambda \langle X, U \rangle + \lambda \langle M, U \rangle - \frac{\mu}{2} \|U\|_F^2 \qquad (30)$$

We set the regularization parameter $\lambda = (\max\{m, n\})^{-0.5}$. We conduct experiment on a synthetic data with $m = n = 100$. To generate the corrupted matrix $O \in \mathbb{R}^{m \times n}$, we first obtain two orthogonal matrices $S_1 \in \mathbb{R}^{m \times k}$ and $S_2 \in \mathbb{R}^{n \times k}$ ($k = 10$) by Gaussian distribution. The low rank matrix $X$ can be calculated by $X = S_1 S_2^\top$. Then we randomly add Gaussian noise to $10\%$ elements of $X$ and obtain the corrupted matrix $O$.

We compare HOPS-D, HOPS-F and PD-HOPS with PD, APG-D and APG-F in our experiments. To make fair comparison, we stop each algorithm when the optimality gap is less than a given $\epsilon$ and count the number of iterations and the running time that each algorithm requires. We set $\epsilon = 10^{-4}, 10^{-5}$ for linear classification problem, and $\epsilon = 10^{-3}, 10^{-4}$ for other two problems. For APG, we use the backtracking trick to tune $L_\mu$. For HOPS, we tune the number of iterations $t$ in each epoch among several values in $\{10, 50, 100, 150, 200, 250, 300, 350, 400, 500, 1000\}$ and the parameter $b$ among $\{1.2, 2, 2.5, 3, 3.5, 4, 5, 10, 25\}$, and report the best results. We also tune the values of parameters $\sigma$ and $\tau$ and report the best results for PD.

## Footnotes

[1]This could be a general norm.