[Reviews · NeurIPS 2016]

Reviewer 1

Summary

The submission considers algorithms for solving a specific class of optimization problems, namely min_{x in Omega_1} F(x), where F(x) = max_{u in Omega_2} \langle Ax, u \rangle - phi(u) + g(x). Here, g is convex, Omega_1 is closed and convex, Omega_2 is closed, convex, and bounded, and the set of optimal solutions Omega_* \subset Omega_1 is convex, compact, and non-empty. The submission also assumes a proximal mapping for g can be computed efficiently. The above framework is apparently general enough to capture a number of applications, including various natural regularized empirical loss minimization problems that arise in machine learning. Classic work of Nesterov combined a smooth approximation technique with accelerated proximal gradient descent to converge to a solution with epsilon of optimal in O(1/epsilon) iterations. Roughly, the smoothing parameter mu controls both the quality of the smooth approximation to the original problem (smaller mu means a better approximation), and the speed of convergence (smaller mu means slower convergence). Choosing mu to balance these costs yields an eps-accurate solution in O(1/eps) iterations. The submission shows that if F satisfies a "Local Error Bound" (LEB), then one can obtain faster convergence. Roughly, an LEB guarantees that the distance of any x from an optimal solution falls like a constant power of the distance of the value F(x) from the optimal value F(x^*). The key idea is to use Nesterov's approach, but the start with a large mu, and gradually decrease it over the course of the execution. The point is that one does not need a good approximation to the original problem at the start (since one is far from an optimal solution then anyway). But as a good solution is approached, one can afford to drop mu to get a better smooth approximation, without killing the convergence rate. A primal-dual version of the algorithm that does not require hand-tuning of parameters is also described (mostly in the supplement). Applications to specific classes of problems are discussed, and some experimental results from 3 different application domains are provided. The experiments show significantly improved iteration complexity for small values of epsilon, compared to basic Accelerated Proximal Gradient Descent, and a first-order primal-dual method.

Qualitative Assessment

This seems like a nice contribution, both in terms of the analysis (which is clean, intuitive, and apparently the first for this kind of smoothing algorithm), and the concrete efficiency as demonstrated in the experiments. (However, I am not knowledgeable enough about the area to know for certain that the algorithms experimentally compared against truly state of the art). The class of problems considered capture a variety of important applications, and the theoretical results do seem to generalize or asymptotically improve on several directly comparable prior works. For applications where theta=1, such as regularized empirical loss minimization with L_1-norm or L_{infty}-norm as the regularizer and a non-smooth loss function like hinge or absolute loss, the submission's algorithm will converge in O(log(epsilon_0/epsilon)) iterations, while the submission suggests that prior work would require O(1/epsilon) iterations to converge. Is this accurate? If so, then this is an exponential improvement on prior work, which seems rather impressive, and perhaps the submission should emphasize this point more. It would be nice to see how the proposed algorithm fares against prior art for larger values of epsilon. My understanding from the experiments is that the advantage of the present algorithm is seen only at small values of epsilon. But if this is the case, why not use the other methods to quickly find a reasonably good solution, and then run the new algorithm starting from said solution? Smaller questions: *The submission states that if L_mu is difficult to compute, one can use the "backtracking trick". Was this necessary in any of the applications considered in the experiments? *Since the proposed primal-dual method requires operating on the dual as well as the primal, it is perhaps surprising that the experimental wall time for this version of the algorithm was best overall. Did the reported wall time account for both the time to update the primal solution and the dual?

Confidence in this Review

2-Confident (read it all; understood it all reasonably well)


Reviewer 2

Summary

Review of "Homotopy smoothing for non-convex problems with lower complexity than O(1/e)" The paper explains a new homotopy smoothing algorithm for solving non-smooth optimization problems. The introduction and related work sections explain the problem setting and specific contribution. Whereas the Nestorov method uses a fixed smoothing parameter for all iterations, the proposed HOPS method changes the smoothing parameter to achieve speedups in early iterations, and better approximations of the non-smooth objective at later iterations. The paper provides the first theoretical guarantee of the complexity of the homotopy smoothing method. Sections 3 and 4 explain the homotopy smoothing method. The text in Section 4.2 is especially helpful for understanding the main idea of the homotopy method. Figure 1 is also helpful for understanding the local sharpness theta parameter which appears in the local error bounds. It is nice to see that in Section 4.4 the duality gap is used to monitor the accuracy of the solution. Section 5 gives some experimental results, and Section 6 is a brief conclusion.

Qualitative Assessment

In general the paper is a nice contribution to the optimization literature, and gives insight into why the homotopy smoothing method works. The main weak point of the paper is the experimental section. Without some measure of error (standard deviation, confidence interval), it is difficult to believe statements about significant differences between the speed of the algorithms. Also, I was expecting to see some experiment/figure that shows how the speed of convergence is affected by the choice of homotopy smoothing parameter (b). One end with b=1 (no updates to the smoothing parameter) it should be relatively slow (same as APG), and then it should speed up as b increases. Right? Minor comments: - In the experiments, what did you use for the b parameter of the HOPS algorithm? Does it matter? - what does "the optimal value is obtained by running PD with a sufficiently large number of iterations such that the duality gap is very small" mean? Why do you need to know the optimal value? You should be able to compute the duality gap without having to know the optimal value.

Confidence in this Review

2-Confident (read it all; understood it all reasonably well)


Reviewer 3

Summary

The paper addresses the composite optimization problem: minimize F = f + g, where neither g nor f need be smooth. The current paper assumes that g has an efficient prox operator, and focuses on f with the rather general structure: f(x) = max_u(< Ax,u > - \phi(u)) The paper proposes to repeatedly approximately minimize versions of f smoothed by amounts that decrease exponentially. The main result Thm 5 shows that when F fulfills a \theta Local Error Bound (LEB) condition (for theta in (0,1]), the number of iterations needed to optimize smoothed versions is O(eps^(theta-1)). For theta = 1 (absolute value, hinge loss...), this leads to convergence linear in total iterations; for theta = 0.5 (e.g., strongly convex functions), this leads to 1/t^2 rates. The paper discusses some different conditions that result in LEB conditions, including KL property, and hinge loss ERM as in SVM. The paper shows some experimental results comparing to other first order methods.

Qualitative Assessment

Technical quality: - The experiments are interesting and well done, except for a significant point reported below. - The proof Thm 5 is explained clearly in the supp. mat., and the aspects in the paper itself (Thm, definitions, examples) are sound well written. Novelty: I am not sufficiently familiar with the exact literature, so am assuming the authors claims hold. In this case, the new rates under the proposed setting are new and non-trivial. It is worth noting that there exist previous work on gradual smoothing for faster rates, such as "Optimal Black-Box Reductions Between Optimization Objectives" despite it being quite different in assumptions and results. Impact: The paper shows how LEB and KL can enable faster than typical rates for problems where f,g are both non-smooth, which has potential to spur further research into methods adaptive to such data dependent parameters. It would be even stronger: - If it considered in detail the computational complexity for ERM problems, e.g., by replacing APG by an SVRG variant. - If it focused on representative performance. The experiments report the best HOPS performance among several arbitrary seeming settings of t, but this is not a practical way to use the algorithm. If PD-HOPS fully avoids this, it should be the focus, and we should see its downsides as well (in the paper, not just the supp. mat.). Clarity and presentation: The sketch of Thm 5 is not clear enough to be useful. The most salient point is the use of x^dagger_s-1,eps and its interaction with Lemma 1, but the induction is neither clear there nor that interesting. l.231: Typo: min_{x\in\RR^}F(x) is a number, does not have an epigraph, remove the min.

Confidence in this Review

2-Confident (read it all; understood it all reasonably well)


Reviewer 4

Summary

This paper shows how to run Nesterov’s smooth minimization of non-smooth functions technique in phases, with increasingly better strongly convex regularizers for the dual, in order to obtain faster convergence for a class of optimization problems. Improvement follows from the fact that the iterate obtained at the end of a phase can be used to warm start the next phase. This is guaranteed via a simple lemma bounding norm distance to sublevel sets in terms of value distance. Together with a local error bound condition, better convergence follows for a set of problems.

Qualitative Assessment

I find this idea really cute. It is a very intuitive thing to do; however, for it seems that we lacked rigorous analyses for such a “homotopy” method. The paper is a pleasure to read, and it introduces the reader very gently to the topic, by first presenting Nesterov’s smoothing trick. It would be interesting to write down what is the precise iteration bound for optimizing Hoffman’s bound and for cone programming (linear convergence for these problems sounds very exciting; I don’t know if it is new or not, and it would be nice to mention whether anything similar results via first order methods were previously known). ### update ### I still like the paper, so I maintain my scores. A quick Google search shows that this is not quite the first claim of a first order method for LP with linear convergence. It's not clear to me how various parameters of the problem affect the iteration count, so I suggest carefully analyzing those instances before claiming something in that direction.

Confidence in this Review

2-Confident (read it all; understood it all reasonably well)


Reviewer 5

Summary

The paper considers a non-smooth optimization problem with a composite objective, the first part of which is given by an explicit max structure. By combining Nesterov's smoothing and acceleration with Yang and Lin's recent result [23], the authors propose a homotopy method which converges at a rate potentially faster than O(1/\epsilon). The actual rate depends on the error bound parameter \theta, and the paper gives some examples of functions for which the error bound holds.

Qualitative Assessment

The paper is well written and easy to read. It uses familiar ideas and proof techniques from convex optimization, but presents them in an elegant way, resulting in what seems to be the first analysis of Nesterov smoothing and acceleration that gives a faster rate than O(1/\epsilon) for non-smooth functions. The result relies on the local error bound property. The authors make a connection between this class and other properties of functions commonly studied in convex optimization, such as local strong convexity, and the Kurdyka-Lojasiewicz property. I would have liked to see these and other connections further discussed, and I encourage the authors to compare their work to some other related works such as [1,2]. Another concern is how knowledge of the constants c and theta affects the performance of the algorithm. Prior knowledge of these constants seems essential for applying the algorithm, and the authors should discuss how they can be estimated, or in which applications it is reasonable to assume knowledge of these constants. Section 4.4 hints that the primal-dual version of the algorithm can solve this problem, but this is not convincing and the discussion should be significantly expanded. Below are additional comments that I encourage the authors to consider: - The comment on line 75 (it can be made sharp by some transformation) is ambiguous. To clarify, you should refer to Definition 4 and the following remark. - The proofs seem to be heavily inspired from [23]. This should be acknowledged. - The notation for the dual norm is not standard, is there a particular reason for this? - Equation (11) can be improved by moving the quantifier to the beginning. - In applications, discuss whether the constant c is typically dimension independent. - Line 231, min_x F(x) has a polyhedral epigraph: this statement does not make sense to me, since min_x F(x) is a scalar, and has no epigraph. - The example Line 240-242: this function is not necessarily locally strongly convex, if x^* - x \in Null(A) then ||x - x^*|| > 0 but F(x) - F(x^*) = 0. Perhaps the authors meant that this function satisfies condition (10) - Define what o-minimal structure means (line 252) [1] Zhi-Quan Luo and Paul Tseng, Error bounds and convergence analysis of feasible descent methods: a general approach. Annals of Operations Research, 1993. [2] B. T. Polyak. Gradient methods for minimizing functionals. Zhurnal Vychislitel’noi Matematiki i Matematicheskoi Fiziki, 3(4):643–653, 1963

Confidence in this Review

3-Expert (read the paper in detail, know the area, quite certain of my opinion)


Reviewer 6

Summary

The authors propose a new optimization algorithm based on a homotopy smoothing, which requires O(1/eps^(1-\theta)\log(1/\eps)) iterations to achieve an eps gap towards the optimal objective value.

Qualitative Assessment

Based on my understanding of existing literature, most of the results in this paper are known or straightforward generalization of known results. For example: (1) Zhang et al. (2012) has shown that for sigma-strongly convex function, an iteration complexity of O(eps^{-1/2}sigma^{-1}). The analysis in Zhang et al. (2012) can be further extended to locally strongly convex optimization, which is identical to the result in this paper on Line 237 of Page 7. Zhang et al. (2012) Smoothing Multivariate Performance Measures. JMLR --http://www.jmlr.org/papers/volume13/zhang12d/zhang12d.pdf (2) The KL condition on Line 243 of Page 7 is also known as the Polyak-Lojasiewicz, or gradient dominated condition. There have been quite a few papers, which used the PL condition to relax the strong convexity. I suggest the authors to give more concrete justifications of the significance of the proposed algorithm.

Confidence in this Review

2-Confident (read it all; understood it all reasonably well)